# Efficient Gradient Computation for Structured Output Learning with Rational and Tropical Losses

**Corinna Cortes**
Google Research
New York, NY 10011
corinna@google.com

**Vitaly Kuznetsov**
Google Research
New York, NY 10011
vitalyk@google.com

**Mehryar Mohri**
Courant Institute and Google Research
New York, NY 10012
mohri@cims.nyu.edu

**Dmitry Storcheus**
Courant Institute and Google Research
New York, NY 10012
dstorcheus@google.com

**Scott Yang**[*]
D. E. Shaw and Co.
New York, NY 10036
yangs@cims.nyu.edu

## Abstract

Many structured prediction problems admit a natural loss function for evaluation such as the edit-distance or $n$-gram loss. However, existing learning algorithms are typically designed to optimize alternative objectives such as the cross-entropy. This is because a naïve implementation of the natural loss functions often results in intractable gradient computations. In this paper, we design efficient gradient computation algorithms for two broad families of structured prediction loss functions: rational and tropical losses. These families include as special cases the $n$-gram loss, the edit-distance loss, and many other loss functions commonly used in natural language processing and computational biology tasks that are based on sequence similarity measures. Our algorithms make use of weighted automata and graph operations over appropriate semirings to design efficient solutions. They facilitate efficient gradient computation and hence enable one to train learning models such as neural networks with complex structured losses.

## 1 Introduction

Many important machine learning tasks are instances of structured prediction problems. These are learning problems where the output labels admit some structure that is important to take into account both for statistical and computational reasons. Structured prediction problems include most natural language processing tasks, such as pronunciation modeling, part-of-speech tagging, context-free parsing, dependency parsing, machine translation, speech recognition, where the output labels are sequences of phonemes, part-of-speech tags, words, parse trees, or acyclic graphs, as well as other sequence modeling tasks in computational biology. They also include a variety of problems in computer vision such as image segmentation, feature detection, object recognition, motion estimation, computational photography and many others.

Several algorithms have been designed in the past for structured prediction tasks, including Conditional Random Fields (CRFs) (Lafferty et al., 2001; Gimpel and Smith, 2010), StructSVMs (Tsochantaridis et al., 2005), Maximum-Margin Markov Networks (M3N) (Taskar et al., 2003), kernel-regression-based algorithms (Cortes et al., 2007), and search-based methods (Daumé III et al., 2009; Doppa et al., 2014; Lam et al., 2015; Chang et al., 2015; Ross et al., 2011). More recently, deep learning techniques have been designed for many structured prediction tasks, including part-of-speech

---

[*]Work done at the Courant Institute of Mathematical Sciences.

tagging (Jurafsky and Martin, 2009; Vinyals et al., 2015a), named-entity recognition (Nadeau and Sekine, 2007), machine translation (Zhang et al., 2008; Wu et al., 2016), image segmentation (Lucchi et al., 2013), and image annotation (Vinyals et al., 2015b).

Many of these algorithms have been successfully used with specific loss functions such as the Hamming loss. Their use has been also extended to multivariate performance measures such as Precision/Recall or $F_1$-score (Joachims, 2005), which depend on predictions on all training points. However, the natural loss function relevant to a structured prediction task, which may be the $n$-gram loss, the edit-distance loss, or some sequence similarity-based loss, is otherwise often ignored. Instead, an alternative measure such as the cross-entropy is used. This is typically due to computational efficiency reasons: a key subroutine within the main optimization such as one requiring to determine the most violating constraint may be computationally intractable, the gradient may not admit a closed-form or may seem difficult to compute, as it may involve sums over a number of terms exponential in the size of the input alphabet, with each term in itself being a large non-trivial computational task.

Several techniques have been suggested in the past to address this issue. They include Minimum Risk Training (MRT) (Och, 2003; Shen et al., 2016), which seeks to optimize the natural objective directly but relies on sampling or focusing on only the top-$n$ structured outputs to make the problem computationally tractable. REINFORCE-based methods (Ranzato et al., 2015; Wu et al., 2016) also seek to optimize the natural loss function by defining an unbiased stochastic estimate of the objective, thereby making the problem computationally tractable. While these publications have demonstrated that training directly with the natural loss function yields better results than using a naïve loss function, their solutions naturally suffer from issues such as high variance in the gradient estimate, in the case of sampling, or bias in the case of top-$n$. Moreover, REINFORCE methods often have to feed the ground-truth at training time, which is inconsistent with the underlying theory.

Another technique has consisted of designing computationally more tractable surrogate loss functions closer to the natural loss function (Ranjbar et al., 2013; Eban et al., 2017). These publications also report improved performance using an objective closer to the natural loss, while admitting the inherent issue of not optimizing the desired metric. McAllester et al. (2010) propose a perceptron-like update in the special case of linear models in structured prediction problems, which avoids the use of surrogate losses. However, while they show that direct loss minimization admits some asymptotic statistical benefits, each update in their work requires solving an $\text{argmax}$ problem for which the authors do not give an algorithm and that is known to be computationally hard in general, particularly for non-additive losses.

This paper is strongly motivated by much of this previous work, which reports empirical benefits for using the natural loss associated to the task. We present efficient gradient computation algorithms for two broad families of structured prediction loss functions: rational and tropical losses. These families include as special cases the $n$-gram loss, the edit-distance loss, and many other loss functions commonly used in natural language processing and computational biology tasks that are based on sequence similarity measures. Our algorithms make use of weighted automata and graph operations over appropriate semirings to design efficient solutions that circumvent the naïve computation of exponentially sized sums in gradient formula.

Our algorithms enable one to train learning models such as neural networks with complex structured losses. When combined with the recent developments in automatic differentiation, e.g. CNTK (Seide and Agarwal, 2016), MXNet (Chen et al., 2015), PyTorch (Paszke et al., 2017), and TensorFlow (Abadi et al., 2016), they can be used to train structured prediction models such as neural networks with the natural loss of the task. In particular, the use of our techniques for the top layer of neural network models can further accelerate progress in end-to-end training (Amodei et al., 2016; Graves and Jaitly, 2014; Wu et al., 2016).

For problems with limited data, e.g. uncommon languages or some biological problems, our work overcomes the computational bottleneck, uses the exact loss function, and renders the amount of data available the next hurdle for improved performance. For extremely large-scale problems with more data than can be processed, we further present an approximate truncated shortest-path algorithm that can be used for fast approximate gradient computations of the edit-distance.

The rest of the paper is organized as follows. In Section 2, we briefly describe structured prediction problems and algorithms, discuss their learning objectives, and point out the challenge of gradient computation. Section 3 defines several weighted automata and transducer operations that we use to

design efficient algorithms for gradient-based learning. In Sections 4 and 5, we give general algorithms for computing the gradient of rational and tropical loss functions, respectively. In Section 6, we report the results of experiments verifying the improvement due to using our efficient methods compared to a naïve implementation. Further details regarding weighted automata and transducer operations and training recurrent neural network training with the structured objective are presented in Appendix A and Appendix B.

## 2 Gradient computation in structured prediction

In this section, we introduce the structured prediction learning problem. We start by defining the learning scenario, including the relevant loss functions and features. We then move on to discussing the hypothesis sets and forms of the objection function that are used by many structured prediction algorithms, which leads us to describe the problem of computing their gradients.

### 2.1 Structured prediction learning scenario

We consider the supervised learning setting, in which the learner receives a labeled sample $S = \{(x_1, y_1), \ldots, (x_m, y_m)\}$ drawn i.i.d. from some unknown distribution over $\mathcal{X} \times \mathcal{Y}$, where $\mathcal{X}$ denotes the input space and $\mathcal{Y}$ the output space. In structured prediction, we assume that elements of the output space $\mathcal{Y}$ can be decomposed into possibly overlapping substructures $y = (y^1, \ldots, y^l)$. We further assume that the loss function $\mathsf{L} \colon \mathcal{Y} \times \mathcal{Y} \to \mathbb{R}_+$ can similarly be decomposed along these substructures. Some key examples of loss functions that are relevant to our work are the Hamming loss, the $n$-gram loss and the edit-distance loss.

The *Hamming loss* is defined for all $y = (y^1, \ldots, y^l)$ and $y' = (y'^1, \ldots, y'^l)$ by $\mathsf{L}(y, y') = \frac{1}{l} \sum_{k=1}^{l} 1_{y^k \neq y'^k}$, with $y^k, y'^k \in \mathcal{Y}_k$. The *edit-distance loss* is commonly used in natural language processing (NLP) applications where $\mathcal{Y}$ is a set of sequences defined over a finite alphabet, and the loss function between two sequences $y$ and $y'$ is defined as the minimum cost of a sequence of edit operations, typically insertions, deletions, and substitutions, that transform $y$ into $y'$. The *$n$-gram loss* is defined as the negative inner product (or its logarithm) of the vectors of $n$-gram counts of two sequences. This can serve as an approximation to the BLEU score, which is commonly used in machine translation.

We assume that the learner has access to a feature mapping $\mathbf{\Psi} \colon \mathcal{X} \times \mathcal{Y} \to \mathbb{R}^N$. This mapping can be either a vector of manually designed features, as in the application of the CRF algorithm, or the differentiable output of the penultimate layer of an artificial neural network. In practice, feature mappings that correspond to the inherent structure of the input space $\mathcal{X}$ combined with the structure of $\mathcal{Y}$ can be exploited to derive effective and efficient algorithms. As mentioned previously, a common case in structured prediction is when $\mathcal{Y}$ is a set of sequences of length $l$ over a finite alphabet $\Delta$. This is the setting that we will consider, as other structured prediction problems can often be treated similarly.

We further assume that $\mathbf{\Psi}$ admits a *Markovian property of order $q$*, that is, for any $(x, y) \in \mathcal{X} \times \mathcal{Y}$, $\mathbf{\Psi}(x, y)$ can be decomposed as $\mathbf{\Psi}(x, y) = \sum_{s=1}^{l} \boldsymbol{\psi}(x, y^{s-q+1:s}, s)$, for some position-dependent feature vector function $\boldsymbol{\psi}$ defined over $\mathcal{X} \times \Delta^q \times [l]$, where the shorthand $y^{s:s'} = (y^s, \ldots, y^{s'})$ stands for the substring of $y$ starting at index $s$ and ending at $s'$. For convenience, for $s \leq 0$, we define $y^s$ to be the empty string $\varepsilon$. This Markovian assumption is commonly adopted in structured prediction problems such as NLP (Manning and Schütze, 1999). In particular, it holds for feature mappings that are frequently used in conjunction with the CRF, as well as outputs of a recurrent neural network, reset at the begining of each new input (see Appendix B).

### 2.2 Objective function and gradient computation

The hypothesis set we consider is that of linear functions $h \colon (x, y) \mapsto \mathbf{w} \cdot \mathbf{\Psi}(x, y)$ based on the feature mapping $\mathbf{\Psi}$. The empirical loss $\widehat{R}_S(h) = \frac{1}{m} \sum_{i=1}^{m} \mathsf{L}(h(x_i), y_i)$ associated to a hypothesis $h$ is often not differentiable in structured prediction since the loss function admits discrete values. Taking the expectation over the distribution induced by the log-linear model, as in (Gimpel and Smith, 2010)[Equation 5], does not help resolve this issue, since the method does not result in an upper bound on the empirical loss and does not admit favorable generalization guarantees. Instead, as in the

familiar binary classification scenario, one can resort to upper-bounding the loss with a differentiable (convex) surrogate. For instance, by (Cortes et al., 2016)[Lemma 4], $\widehat{R}_S(h)$ can be upper-bounded by the following objective function:

$$F(\mathbf{w}) = \frac{1}{m} \sum_{i=1}^m \log \left[ \sum_{y \in \mathcal{Y}} e^{\mathsf{L}(y,y_i) - \mathbf{w} \cdot (\mathbf{\Psi}(x_i,y_i) - \mathbf{\Psi}(x_i,y))} \right], \tag{1}$$

which, modulo a regularization term, coincides with the objective function of CRF. Note that this expression has also been presented as the softmax margin (Gimpel and Smith, 2010) and the reward-augmented maximum likelihood (Norouzi et al., 2016). Both of these references demonstrate strong empirical evidence for this choice of objective function (in addition to the theoretical results presented in (Cortes et al., 2016)).

Our focus in this work is on an efficient computation of the gradient of this objective function. Since the computation of the subgradient of the regularization term often does not pose any issues, we will only consider the unregularized part of the objective. For any $\mathbf{w}$ and $i \in [m]$, let $F_i(\mathbf{w})$ denote the contribution of the $i$-th training point to the objective function $F$. A standard gradient descent-based method would sum up all or a subset (mini-batch) of the gradients $\nabla F_i(\mathbf{w})$. As illustrated in (Cortes et al., 2016)[Lemma 15], the gradient $\nabla F_i(\mathbf{w})$ can be expressed as follows at any $\mathbf{w}$:

$$\nabla F_i(\mathbf{w}) = \frac{1}{m} \sum_{s=1}^l \sum_{\mathbf{z} \in \Delta^q} \mathsf{Q}_{\mathbf{w}}(\mathbf{z}, s) \boldsymbol{\psi}(x_i, \mathbf{z}, s) - \frac{\mathbf{\Psi}(x_i, y_i)}{m},$$

where, for all $\mathbf{z} \in \Delta^q$ and $s \in [l]$, $\mathsf{Q}_{\mathbf{w}}(\mathbf{z}, s)$ is defined by

$$\mathsf{Q}_{\mathbf{w}}(\mathbf{z}, s) = \sum_{y \,:\, y^{s-q+1:s} = \mathbf{z}} \frac{e^{\mathsf{L}(y,y_i) + \mathbf{w} \cdot \mathbf{\Psi}(x_i,y)}}{Z_{\mathbf{w}}} \quad \text{and} \quad Z_{\mathbf{w}} = \sum_{y \in \mathcal{Y}} e^{\mathsf{L}(y,y_i) + \mathbf{w} \cdot \mathbf{\Psi}(x_i,y)}.$$

The bottleneck in the gradient computation is the evaluation of $\mathsf{Q}_{\mathbf{w}}(\mathbf{z}, s)$, for all $\mathbf{z} \in \Delta^q$ and $s \in [l]$. There are $l|\Delta|^q$ such terms and each term $\mathsf{Q}_{\mathbf{w}}(\mathbf{z}, s)$ is defined by a sum over the $|\Delta|^{l-q}$ sequences $y$ of length $l$ with a fixed substring $\mathbf{z}$ of length $q$. A straightforward computation of these terms following their definition would therefore be computationally expensive. To avoid that computational cost, many existing learning algorithms for structured prediction, including most of those mentioned in the introduction, resort to further approximations and omit the loss $\mathsf{L}$ from the definition of $\mathsf{Q}_{\mathbf{w}}(\mathbf{z}, s)$. Combining that with the Markovian structure of $\mathbf{\Psi}$ can then lead to efficient gradient computations. Of course, the caveat of this approach is that it ignores the key component of the learning problem, namely the loss function.

In what follows, we will present efficient algorithms for the exact computation of the terms $\mathsf{Q}_{\mathbf{w}}(\mathbf{z}, s)$, with their full definition, including the loss function. This leads to an efficient computation of the gradients $\nabla F_i$, which can be used as input to back-propagation algorithms that would enable us to train neural network models with structured prediction losses.

The gradient computation methods we present apply to the Hamming loss, $n$-gram loss, and edit-distance loss, and more generally to two broad families of losses that can be represented by weighted finite-state transducers (WFSTs). This covers many losses based on sequence similarity measures that are used in NLP and computational biology applications (Cortes et al., 2004; Schölkopf et al., 2004).

We briefly describe the WFST operations relevant to our solutions in the following section and provide an example of how the edit-distance loss can be represented with a WFST in Section 5.

## 3   Weighted automata and transducers

Weighted finite automata (WFA) and weighted finite-state transducers (WFST) are fundamental concepts and representations widely used in computer science (Mohri, 2009). We will use WFAs and WFSTs to devise algorithms that efficiently compute gradients of structured prediction objectives. This section introduces some standard concepts and notation for WFAs and WFSTs. We provide additional details in Appendix A. For a more comprehensive treatment of these topics, we refer the reader to (Mohri, 2009).

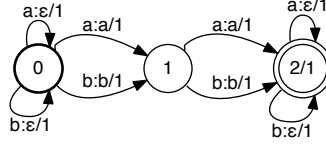

**Figure 1:** Bigram transducer $\mathcal{T}_{\text{bigram}}$ over the semiring $(\mathbb{R}_+ \cup \{+\infty\}, +, \times, 0, 1)$ for the alphabet $\Delta = \{a, b\}$. The weight of each transition (or that of a final state) is indicated after the slash separator. For example, for any string $y$ and bigram $\mathbf{u}$, $\mathcal{T}_{\text{bigram}}(y, \mathbf{u})$ is equal to the number of occurrences of $\mathbf{u}$ in $y$ (Cortes et al., 2015).

*Definition.* A *weighted finite-state transducer* $\mathcal{T}$ over a semiring $(\mathbb{S}, \oplus, \otimes, \overline{0}, \overline{1})$ is an 8-tuple $(\Sigma, \Delta, Q, I, F, E, \lambda, \rho)$ where $\Sigma$ is a finite input alphabet, $\Delta$ is a finite output alphabet, $Q$ is a finite set of states, $I \subseteq Q$ is the set of initial states, $F \subseteq Q$ is the set of final states, $E$ is a finite multiset of transitions, which are elements of $Q \times (\Sigma \cup \{\epsilon\}) \times (\Delta \cup \{\epsilon\}) \times \mathbb{S} \times Q$, $\lambda \colon I \to \mathbb{S}$ is an initial weight function, and $\rho \colon F \to \mathbb{S}$ is a final weight function. A *weighted finite automaton* is a weighted finite-state transducer where the input and output labels are the same. See Figures 1 and 3 for some examples.

For many operations to be well defined, the weights of a WFST must belong to a semiring $(\mathbb{S}, \oplus, \otimes, \overline{0}, \overline{1})$. We provide a formal definition of a semiring in Appendix A. In this work, we consider two semirings: the probability semiring $(\mathbb{R}_+ \cup \{+\infty\}, +, \times, 0, 1)$ and the tropical semiring $(\mathbb{R} \cup \{-\infty, +\infty\}, \min, +, +\infty, 0)$. The $\otimes$-operation is used to compute the weight of a path by $\otimes$-multiplying the weights of the transitions along that path. The $\oplus$-operation is used to compute the weight of a pair of input and output strings $(x, y)$ by $\oplus$-summing the weights of the paths labeled with $(x, y)$. We denote this weight by $\mathcal{T}(x, y)$.

As shown in Sections 4 and 5, in many useful cases, we can reduce the computation of the loss function $\mathsf{L}(y, y')$ between two strings $y$ and $y'$, along with the gradient of the corresponding objective described in (1), to that of the $\oplus$-sum of the weights of all paths labeled by $y{:}y'$ in a suitably defined transducer over either the probability or tropical semiring. We will use the following standard WFST operations to construct these transducers: inverse ($\mathcal{T}^{-1}$), projection ($\Pi(\mathcal{T})$), composition ($\mathcal{T}_1 \circ \mathcal{T}_2$), and determinization ($\text{Det}(\mathcal{A})$). The definitions of these operations are given in Appendix A.

## 4    An efficient algorithm for the gradient computation of rational losses

As discussed in Section 2, computing $\mathsf{Q}_\mathbf{w}(\mathbf{z}, s)$ is the main bottleneck in the gradient computation. In this section, we give an efficient algorithm for computing $\mathsf{Q}_\mathbf{w}(\mathbf{z}, s)$ that works for an arbitrary *rational loss*, which includes as a special case the $n$-gram loss and other sequence similarity-based losses. We first present the definition of a rational loss and show how the $n$-gram loss can be encoded as a specific rational loss. Then, we present our gradient computation algorithm.

Let $(\mathbb{R}_+ \cup \{+\infty\}, +, \times, 0, 1)$ be the *probability semiring* and let $\mathcal{U}$ be a WFST over the probability semiring admitting $\Delta$ as both the input and output alphabet. Then, following (Cortes et al., 2015), the *rational loss* associated to $\mathcal{U}$ is the function $L_\mathcal{U} \colon \Delta^* \times \Delta^* \to \mathbb{R} \cup \{-\infty, +\infty\}$ defined for all $y, y' \in \Delta^*$ by $L_\mathcal{U}(y, y') = -\log\big(\mathcal{U}(y, y')\big)$. As an example, the $n$-gram loss of $y$ and $y'$ is the negative logarithm of the inner product of the vectors of $n$-gram counts of $y$ and $y'$. The WFST $\mathcal{U}_{n\text{-gram}}$ of an $n$-gram loss is obtained by composing a weighted transducer $\mathcal{T}_{n\text{-gram}}$ giving the $n$-gram counts with its inverse $\mathcal{T}_{n\text{-gram}}^{-1}$, that is the transducer derived from $\mathcal{T}_{n\text{-gram}}$ by swapping input and output labels for each transition. As an example, Figure 1 shows the WFST $\mathcal{T}_{\text{bigram}}$ for bigrams.

To compute $\mathsf{Q}_\mathbf{w}(\mathbf{z}, s)$ for a rational loss, recall that

$$\mathsf{Q}_\mathbf{w}(\mathbf{z}, s) \propto \sum_{y \colon y^{s-q+1:s} = \mathbf{z}} e^{L_\mathcal{U}(y, y_i) + \mathbf{w} \cdot \mathbf{\Psi}(x_i, y)}.$$

Thus, we will design two WFAs, $\mathcal{A}$ and $\mathcal{B}$, such that $\mathcal{A}(y) = e^{\mathbf{w} \cdot \mathbf{\Psi}(x_i, y)}$, $\mathcal{B}(y) = e^{L_\mathcal{U}(y, y_i)}$, and their composition $\mathcal{C}(y) = (\mathcal{A} \circ \mathcal{B})(y) = e^{L_\mathcal{U}(y, y_i) + \mathbf{w} \cdot \mathbf{\Psi}(x_i, y)}$. To compute $\mathsf{Q}_\mathbf{w}$ from $\mathcal{C}$, we will need to sum up the weights of all paths labeled with some substring $\mathbf{z}$, which we will achieve by treating this as a flow computation problem.

The pseudocode of our algorithm for computing the key terms $\mathsf{Q}_\mathbf{w}(\mathbf{z}, s)$ for a rational loss is given in Figure 2(a).

GRAD-RATIONAL$(x_i, y_i, \mathbf{w})$

1   $\overline{\mathcal{Y}} \leftarrow$ WFA accepting any $y \in \Delta^l$.
2   $\mathcal{Y}_i \leftarrow$ WFA accepting $y_i$.
3   $\mathcal{M} \leftarrow \Pi_1(\overline{\mathcal{Y}} \circ \mathcal{U} \circ \mathcal{Y}_i)$
4   $\mathcal{M} \leftarrow \text{Det}(\mathcal{M})$
5   $\mathcal{B} \leftarrow \text{INVERSEWEIGHTS}(\mathcal{M})$
6   $\mathcal{C} \leftarrow \mathcal{A} \circ \mathcal{B}$
7   $\alpha \leftarrow \text{DISTFROMINITIAL}(\mathcal{C}, (+, \times))$
8   $\beta \leftarrow \text{DISTTOFINAL}(\mathcal{C}, (+, \times))$
9   $Z_{\mathbf{w}} \leftarrow \beta(I_{\mathcal{C}}) \;\triangleright\; I_{\mathcal{C}}$ initial state of $\mathcal{C}$
10  $\mathbf{for}\ (\mathbf{z}, s) \in \Delta^q \times [l]\ \mathbf{do}$
11     $Q_{\mathbf{w}}(\mathbf{z}, s) \leftarrow \displaystyle\sum_{e \in E_{\mathbf{z},s}} \alpha(\underline{e}) \times \omega(e) \times \beta(\overline{e})$
12     $Q_{\mathbf{w}}(\mathbf{z}, s) \leftarrow Q_{\mathbf{w}}(\mathbf{z}, s)/Z_{\mathbf{w}}$

(a)

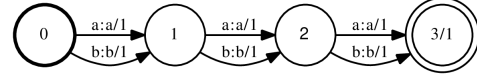

(b)

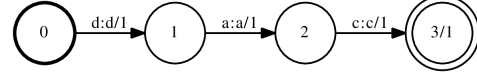

(c)

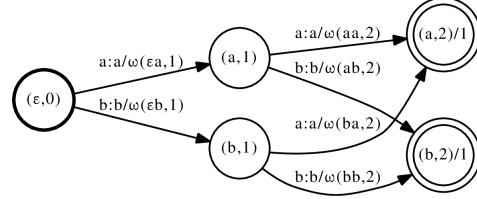

(d)

**Figure 2:** (a) Efficient computation of the key terms of the structured gradient for the rational loss. For each transition $e \in E_{\mathbf{z},s}$, we denote its origin by $\underline{e}$, destination by $\overline{e}$ and weight by $\omega(e)$. (b) Illustration of the WFA $\overline{\mathcal{Y}}$ for $\Delta = \{a, b\}$ and $l = 3$. (c) Illustration of the WFA $\mathcal{Y}_i$ representing string $dac$. (d) Illustration of WFA $\mathcal{A}$ for $q = 2$, alphabet $\Delta = (a, b)$ and string length $l = 2$. For example, the transition from state $(a, 1)$ to state $(b, 2)$ has the label $b$ and weight $\omega(ab, 2) = e^{\mathbf{w} \cdot \boldsymbol{\psi}(x_i, ab, 2)}$.

**Design of $\mathcal{A}$.** We want to design a determnistic WFA $\mathcal{A}$ such that

$$\mathcal{A}(y) = e^{\mathbf{w} \cdot \boldsymbol{\Psi}(x_i, y)} = \prod_{t=1}^{l} e^{\mathbf{w} \cdot \boldsymbol{\psi}(x_i, y^{t-q+1:t}, t)}.$$

To accomplish this task, let $\mathcal{A}$ be a WFA with the following set of states $Q_{\mathcal{A}} = \Big\{ (y^{t-q+1:t}, t) \colon y \in \Delta^l, t = 0, \ldots, l \Big\}$, with $I_{\mathcal{A}} = (\varepsilon, 0)$ its single initial state, $F_{\mathcal{A}} = \{ (y^{l-q+1:l}, l) \colon y \in \Delta^l \}$ its set of final states, and with a transition from state $(y^{t-q+1:t-1}, t-1)$ to state $(y^{t-q+2:t-1} b, t)$ with label $b$ and weight $\omega(y^{t-q+1:t-1} b, t) = e^{\mathbf{w} \cdot \boldsymbol{\psi}(x_i, y^{t-q+1:t-1} b, t)}$, that is, the following set of transitions:

$$E_{\mathcal{A}} = \Big\{ \big( (y^{t-q+1:t-1}, t-1), b, \omega(y^{t-q+1:t-1} b, t), (y^{t-q+2:t-1} b, t) \big) \colon y \in \Delta^l, b \in \Delta, t \in [l] \Big\}.$$

Figure 2(d) illustrates this construction in the case $q = 2$. Note that the WFA $\mathcal{A}$ is deterministic by construction. Since the weight of a path in $\mathcal{A}$ is obtained by multiplying the transition weights along the path, $\mathcal{A}(y)$ computes the desired quantity.

**Design of $\mathcal{B}$.** We now design a deterministic WFA $\mathcal{B}$ which associates to each sequence $y \in \Delta^l$ the exponential of the loss $e^{L_u(y, y_i)} = 1/\mathcal{U}(y, y_i)$. Let $\overline{\mathcal{Y}}$ denote a WFA over the probability semiring accepting the set of all strings of length $l$ with weight one and let $\mathcal{Y}_i$ denote the WFA accepting only $y_i$ with weight one. Figures 2(b) and 2(c) illustrate the constructions of $\overline{\mathcal{Y}}$ and $\mathcal{Y}_i$ in some simple cases.[2] We first use the composition operation for weighted automata and transducers. Then, we use the projection operation on the input, which we denote by $\Pi_1$, to compute the following WFA: $\mathcal{M} = \Pi_1(\overline{\mathcal{Y}} \circ \mathcal{U} \circ \mathcal{Y}_i)$. Recalling that $\overline{\mathcal{Y}}(y) = \mathcal{Y}_i(y_i) = 1$ by construction and applying the definition of WFST composition, we observe that for any $y \in \Delta^l$

$$\mathcal{M}(y) = (\overline{\mathcal{Y}} \circ \mathcal{U} \circ \mathcal{Y}_i)(y, y_i) = \sum_{\mathbf{z}=y, \mathbf{z}'=y_i} \overline{\mathcal{Y}}(\mathbf{z}) \mathcal{U}(\mathbf{z}, \mathbf{z}') \mathcal{Y}_i(\mathbf{z}') = \overline{\mathcal{Y}}(y) \mathcal{U}(y, y_i) \mathcal{Y}_i(y_i) = \mathcal{U}(y, y_i). \quad (2)$$

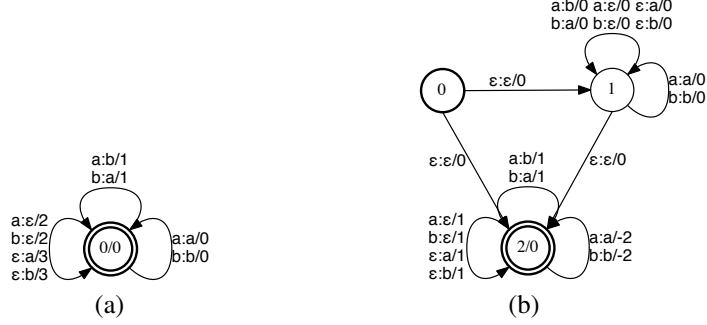

(a)                                                    (b)

**Figure 3:** (a) Edit-distance transducer $\mathcal{U}_{\text{edit}}$ over the tropical semiring, in the case where the substitution cost is 1, the deletion cost 2, the insertion cost 3, and the alphabet $\Delta = \{a, b\}$. (b) Smith-Waterman transducer $\mathcal{U}_{\text{Smith-Waterman}}$ over the tropical semiring, in the case where the substitution, deletion and insertion costs are 1, and where the matching cost is $-2$, for the alphabet $\Delta = \{a, b\}$.

Next, we can apply weighted determinization (Mohri, 1997) to compute a deterministic WFA equivalent to $\mathcal{M}$, denoted by $\text{Det}(\mathcal{M})$. By (Cortes et al., 2015)[Theorem 3], $\text{Det}(\mathcal{M})$ can be computed in polynomial time. Since $\text{Det}(\mathcal{M})$ is deterministic and by construction accepts precisely the set of strings $y \in \Delta^l$, it admits a unique accepting path labeled with $y$ whose weight is $\text{Det}(\mathcal{M})(y) = \mathcal{M}(y) = \mathcal{U}(y, y_i)$. The weight of that accepting path is obtained by multiplying the weights of its transitions and that of the final state. Let $\mathcal{B}$ be the WFA derived from $\text{Det}(\mathcal{M})$ by replacing each transition weight or final weight $u$ by its inverse $\frac{1}{u}$. Then, by construction, for any $y \in \Delta^l$, we have $\mathcal{B}(y) = \frac{1}{\mathcal{U}(y, y_i)}$.

**Combining $\mathcal{A}$ and $\mathcal{B}$.** Now consider the WFA $\mathcal{C} = \mathcal{A} \circ \mathcal{B}$, the composition of $\mathcal{A}$ and $\mathcal{B}$. $\mathcal{C}$ is deterministic since both $\mathcal{A}$ and $\mathcal{B}$ are deterministic. Moreover, $\mathcal{C}$ can be computed in time $O(|\mathcal{A}||\mathcal{B}|)$. By definition, for all $y \in \Delta^l$,

$$\mathcal{C}(y) = \mathcal{A}(y) \times \mathcal{B}(y) = \prod_{t=1}^{l} e^{\mathbf{w} \cdot \boldsymbol{\psi}(x_i, y^{t-q+1:t}, t)} \times \frac{1}{\mathcal{U}(y, y_i)} = e^{\mathsf{L}(y, y_i)} \prod_{t=1}^{l} e^{\mathbf{w} \cdot \boldsymbol{\psi}(x_i, y^{t-q+1:t}, t)}. \quad (3)$$

To see how $\mathcal{C}$ can be used to compute $\mathsf{Q}_w(\mathbf{z}, s)$, we note first that the states of $\mathcal{C}$ can be identified with pairs $(q_\mathcal{A}, q_\mathcal{B})$ where $q_\mathcal{A}$ is a state of $\mathcal{A}$, $q_\mathcal{B}$ is a state of $\mathcal{B}$, and the transitions are obtained by matching a transition in $\mathcal{A}$ with one in $\mathcal{B}$. Thus, for any $\mathbf{z} \in \Delta^q$ and $s \in [l]$, let $E_{\mathbf{z},s}$ be the set of transitions of $\mathcal{C}$ constructed by pairing the transition in $\mathcal{A}$ $((\mathbf{z}^{1:q-1}, s-1), z^q, \omega(\mathbf{z}, s), (\mathbf{z}^{2:q}, s))$ with a transition in $\mathcal{B}$:

$$E_{\mathbf{z},s} = \left\{ \left( (q_\mathcal{A}, q_\mathcal{B}), z^q, \omega, (q'_\mathcal{A}, q'_\mathcal{B}) \right) \in E_\mathcal{C} : q_\mathcal{A} = (\mathbf{z}^{1:q-1}, s-1) \right\}. \quad (4)$$

Note that, since $\mathcal{C}$ is deterministic, there can be only one transition leaving a state labeled with $z^q$. Thus, to define $E_{\mathbf{z},s}$, we only needed to specify the origin state of the transitions.

For each transition $e \in E_{\mathbf{z},s}$, we denote its origin by $\underline{e}$, destination by $\overline{e}$ and weight by $\omega(e)$. Then, $\mathsf{Q_w}(\mathbf{z}, s)$ can be computed as $\sum_{e \in E_{\mathbf{z},s}} \alpha(\underline{e}) \times \omega(e) \times \beta(\overline{e})$, where $\alpha(\underline{e})$ is the sum of the weights of all paths from an initial state of $\mathcal{C}$ to $\underline{e}$, and $\beta(\overline{e})$ is the sum of the weights of all paths from $\overline{e}$ to a final state of $\mathcal{C}$. Since $\mathcal{C}$ is acyclic, $\alpha$ and $\beta$ can be computed for all states in linear time in the size of $\mathcal{C}$ using a single-source shortest-distance algorithm over the $(+, \times)$ semiring (Mohri, 2002) or the so-called forward-backward algorithm. We denote these subroutines by DistFromInitial and DistToFinal respectively in the pseudocode. Since $\mathcal{C}$ admits $O(l|\Delta|^q)$ transitions, we can compute all of the quantities $\mathsf{Q_w}(\mathbf{z}, s)$, $s \in [l]$ and $z \in \Delta^q$ and $Z'_\mathbf{w}$, in time $O(l|\Delta|^q)$.

Note that a natural alternative to the weighted transducer methods presented in this work is to consider junction tree type methods for graphical methods. However, weighted transducer techniques typically result in more "compact" representations than graphical model methods, and the computational cost of the former can even be exponentially faster than the best one could achieve using the latter (Poon and Domingos, 2011).

GRAD-TROPICAL$(x_i, y_i, \mathbf{w})$

1  $\overline{\mathcal{Y}} \leftarrow$ WFA accepting any $y \in \Delta^l$.
2  $\mathcal{Y}_i \leftarrow$ WFA accepting $y_i$.
3  $\mathcal{M} \leftarrow \Pi_1(\overline{\mathcal{Y}} \circ \mathcal{U} \circ \mathcal{Y}_i)$
4  $\mathcal{M} \leftarrow \text{Det}(\mathcal{M})$
5  $\mathcal{B} \leftarrow$ EXPONENTIATEWEIGHTS$(\mathcal{M})$
6  $\mathcal{C} \leftarrow \mathcal{A} \circ \mathcal{B}$
7  $\alpha \leftarrow$ DISTFROMINITIAL$(\mathcal{C}, (+, \times))$
8  $\beta \leftarrow$ DISTTOFINAL$(\mathcal{C}, (+, \times))$
9  $Z_{\mathbf{w}} \leftarrow \beta(I_{\mathcal{C}}) \,\triangleright\, I_{\mathcal{C}}$ initial state of $\mathcal{C}$
10  **for** $(\mathbf{z}, s) \in \Delta^q \times [l]$ **do**
11  $\quad \mathsf{Q}_{\mathbf{w}}(\mathbf{z}, s) \leftarrow \sum_{e \in E_{\mathbf{z}, s}} \alpha(\underline{e}) \times \omega(e) \times \beta(\overline{e})$
12  $\quad \mathsf{Q}_{\mathbf{w}}(\mathbf{z}, s) \leftarrow \mathsf{Q}_{\mathbf{w}}(\mathbf{z}, s) / Z_{\mathbf{w}}$

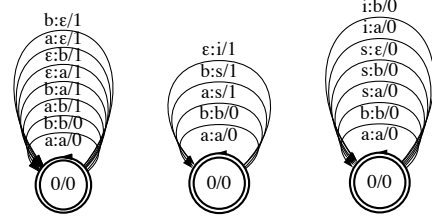

(a)             (b)

**Figure 4:** (a) Efficient computation of the key terms of the structured gradient for the tropical loss. (b) Factoring of the edit-distance transducer. The leftmost figure is the edit-distance weighted transducer $\mathcal{U}_{\text{edit}}$ over alphabet $\Sigma = \{a, b\}$, the center figure is a weighted transducer $\mathcal{T}_1$, and the rightmost figure is a weighted transducer $\mathcal{T}_2$ such that $\mathcal{U}_{\text{edit}} = \mathcal{T}_1 \circ \mathcal{T}_2$.

## 5  An efficient algorithm for the gradient computation of tropical losses

Following the treatment in (Cortes et al., 2015), the *tropical loss* associated to a weighted transducer $\mathcal{U}$ over the tropical semiring is defined as the function $L_{\mathcal{U}}: \Delta^* \times \Delta^* \to \mathbb{R}$ coinciding with $\mathcal{U}$; thus, for all $y, y' \in \Delta^*$, $L_{\mathcal{U}}(y, y') = \mathcal{U}(y, y')$.

For examples of weighted transducers over the tropical semiring, see Figures 3(a) and (b).

Our algorithm for computing $\mathsf{Q}_{\mathbf{w}}(\mathbf{z}, s)$ for a tropical loss, illustrated in Figure 4(a), is similar to our algorithm for a rational loss, with the primary difference being that we exponentiate weights instead of invert them in the WFA $\mathcal{B}$. Specifically, we design $\mathcal{A}$ just as in Section 4, and we design a deterministic WFA $\mathcal{B}$ by first designing $\text{Det}(\mathcal{M})$ as in Section 4 and then deriving $\mathcal{B}$ from $\text{Det}(\mathcal{M})$ by replacing each transition weight or final weight $u$ in $\text{Det}(\mathcal{M})$ by $e^u$. Then by construction, for any $y \in \Delta^l$, $\mathcal{B}(y) = e^{\mathcal{U}(y, y_i)}$. Moreover, composition of $\mathcal{A}$ with $\mathcal{B}$ yields a WFA $\mathcal{C} = \mathcal{A} \circ \mathcal{B}$ such that for all $y \in \Delta^l$,

$$\mathcal{C}(y) = \mathcal{A}(y) \times \mathcal{B}(y) = \prod_{t=1}^{l} e^{\mathbf{w} \cdot \boldsymbol{\psi}(x_i, y^{t-q+1:t}, t)} \times e^{\mathcal{U}(y, y_i)} = e^{\mathsf{L}(y, y_i)} \prod_{t=1}^{l} e^{\mathbf{w} \cdot \boldsymbol{\psi}(x_i, y^{t-q+1:t}, t)}. \quad (5)$$

As an example, the general edit-distance of two sequences $y$ and $y'$ can, as already described, be computed using $\mathcal{U}_{\text{edit}}$ in time $O(|y||y'|)$ (Mohri, 2003). Note that for further computational optimization, $\mathcal{U}_{\text{edit}}$ and $\mathcal{U}_{\text{Smith-Waterman}}$ can be computed on-the-fly as demanded by the composition operation, thereby creating only transitions with alphabet symbols appearing in the strings compared.

In order to achieve optimal dependence on the size of the input alphabet, we can also apply *factoring* to the edit-distance transducer. Figure 4(b) illustrates factoring of the edit-distance transducer over the alphabet $\Sigma = \{a, b\}$, where $s$ is the substitution and deletion symbol and $i$ is the insertion symbol. Note that both $T_1$ and $T_2$ are linear in the size of $\Sigma$, while $\mathcal{U}_{\text{edit}}$ is quadratic in $|\Sigma|$. Furthermore, using on-the-fly composition, for any $\mathcal{Y}_1$ and $\mathcal{Y}_2$, we can first compute $\mathcal{Y}_1 \circ T_1$ and $T_2 \circ \mathcal{Y}_2$ and then compose the result achieving time and space complexity in $O(|\mathcal{Y}_1||\mathcal{Y}_2|)$.

## 6  Experiments

In this section, we present experiments validating both the computational efficiency of our gradient computation methods as well as the learning benefits of training with natural loss functions. The experiments in this section should be treated as a proof of concept. We defer an extensive study of training structured prediction models on large-scale datasets for future work.

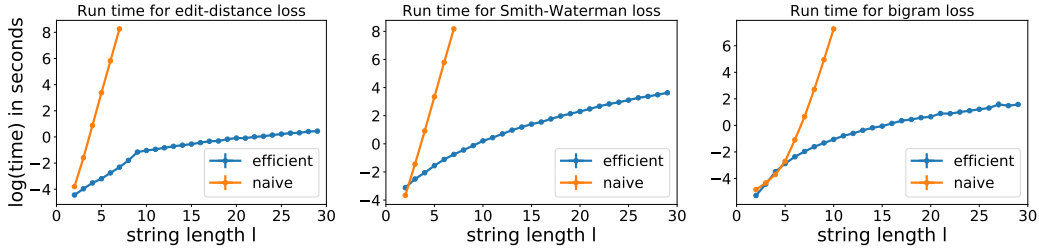

**Figure 5:** Runtime comparison of efficient versus naïve gradient computation methods for edit-distance (a), Smith-Waterman (b) and bigram (c) loss functions. The *naïve* line refers to the average runtime of Grad-Naïve, the *efficient* line refers to Grad-Tropical for edit-distance (a) and Smith-Waterman (b) and Grad-Rational for bigram (c) loss. Naïve computations are shown only up to string length $l = 8$.

For the runtime comparison, we randomly generate an input and output data pair $(x_i, y_i)$, both of a given fixed length, as well as a weight vector $\mathbf{w}$, and we compute $\nabla F_i(\mathbf{w})$ using both the naïve and the outlined efficient methods. As shown in Section 2, the computationally demanding part in the $\nabla F_i(\mathbf{w})$ calculation is evaluating $Q_{\mathbf{w}}(\mathbf{z}, s)$ for all $s \in [l]$ and $\mathbf{z} \in \Delta^q$, while the other terms are generally not problematic to compute. We define a procedure Grad-Naïve (see Figure 6 in the appendix) and compare the average runtimes of Grad-Naïve with that of Grad-Efficient for both rational and tropical losses. The efficient algorithms suggested in this work improve upon the Grad-Naïve runtime by eliminating the explicit loop over $y \in \mathcal{Y}$ and using the weighted automata and transducer operations instead. All the weighted automata and transducer computations required for Grad-Rational and Grad-Tropical are implemented using OpenFST (Allauzen et al., 2007).

More specifically, we define an alphabet $|\Delta| = 10$ and features $\mathbf{\Psi}(x, y)$ as vectors of counts of all 100 possible bigrams. For each string length $l$ from 2 to 30, we draw input pairs $(x_i, y_i) \in \Delta^l \times \Delta^l$ uniformly at random and $\mathbf{w} \in \mathbb{R}^{100}$ according to a standard normal distribution. The average runtimes over 125 random trials are presented in Figure 5 for three loss functions: the edit-distance, the Smith-Waterman distance and the bigram loss. The experiments demonstrate a number of crucial benefits of our efficient gradient computation framework. Note that the Grad-Naïve procedure runtime grows exponentially in $l$, while Grad-Tropical and Grad-Rational exhibit linear dependency on the length of the input strings. In fact, using the threshold pruning as part of determinization can allow one to compute approximate gradient for arbitrarily long input strings. The computational improvement is even more evident for rational losses, in which case the determinization of $\mathcal{M}$ can be achieved in polynomial time (Cortes et al., 2015), thus pruning is not required.

We also provide preliminary learning experiments that illustrate the benefit of learning with a structured loss for a sequence alignment task, compared to training with the cross-entropy loss. The sequence alignment experiment replicates the artificial genome sequence data in (Joachims et al., 2006), where each example consists of native, homolog, and decoy sequences of length 50 and the task is to predict a sequence that is the closest to native in terms of the Smith-Waterman alignment score. The experiment confirms that a model trained with Smith-Waterman distance as the objective shows significantly higher average Smith-Waterman alignment score (and higher accuracy) on a test set compared to a model trained with cross-entropy objective. The cross-entropy model achieved a Smith-Waterman score of 42.73, while the augmented model achieved a score of 44.65 on a test set with a standard deviation of 0.35 averaged over 10 random folds.

# 7 Conclusion

We presented efficient algorithms for computing the gradients of structured prediction models with rational and tropical losses, reporting experimental results confirming both runtime improvement compared to naïve implementations and learning improvement compared to standard methods that settle for using easier-to-optimize losses. We also showed how our approach can be incorporated into the top layer of a neural network, so that it can be used to train end-to-end models in domains including speech recognition, machine translation, and natural language processing. For future work, we plan to run large-scale experiments with neural networks to further demonstrate the benefit of working directly with rational or tropical losses using our efficient computational methods.

**Acknowledgments**

This work was partly funded by NSF CCF-1535987 and NSF IIS-1618662.

## Footnotes

[2]Note that we do not need to explicitly construct $\overline{\mathcal{Y}}$, which could be costly when the alphabet size $\Delta$ is large. Instead, we can create its transitions on-the-fly as demanded by the composition operation. Thus, for the rational kernels commonly used, at most the transitions labeled with the alphabet symbols appearing in $\mathcal{Y}_i$ need to be created.

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
