[Supplementary Material · rtg_supplementary.pdf]

## A Weighted automata and transducers operations

This section provides further details on the concepts of weighted automata and transducers that were introduced in Section 3.

Recall that we have defined a *weighted finite-state transducer* $\mathcal{T}$ over a semiring $(\mathbb{S}, \oplus, \otimes, \overline{0}, \overline{1})$ as an 8-tuple $(\Sigma, \Delta, Q, I, F, E, \lambda, \rho)$, where $\Sigma$ is a finite input alphabet, $\Delta$ is a finite output alphabet, $Q$ is a finite set of states, $I \subseteq Q$ is the set of initial states, $F \subseteq Q$ is the set of final states, $E$ is a finite multiset of transitions, which are elements of $Q \times (\Sigma \cup \{\epsilon\}) \times (\Delta \cup \{\epsilon\}) \times \mathbb{S} \times Q$, $\lambda \colon I \to \mathbb{S}$ is an initial weight function, and $\rho \colon F \to \mathbb{S}$ is a final weight function. Moreover, we defined a *weighted finite automaton* to be a weighted finite-state transducer where the input and output labels are the same.

A tuple $(\mathbb{S}, \oplus, \otimes, \overline{0}, \overline{1})$ is a *semiring* if $(\mathbb{S}, \oplus, \overline{0})$ is a commutative monoid with identity element $\overline{0}$, $(\mathbb{S}, \otimes, \overline{1})$ is a monoid with identity element $\overline{1}$, $\otimes$ distributes over $\oplus$, and $\overline{0}$ is an annihilator for $\otimes$. In other words, a semiring is a ring that may lack negation.

The construction of weighted transducers and automata used in Sections 4 and 5 required the following operations: inverse $(\mathcal{T}^{-1})$, projection $(\Pi(\mathcal{T}))$, composition $(\mathcal{T}_1 \circ \mathcal{T}_2)$, and determinization $(\text{Det}(\mathcal{A}))$. We provide precise definitions of these operations below.

The *inverse* of a WFST $\mathcal{T}$ is denoted by $\mathcal{T}^{-1}$ and defined as the transducer obtained by swapping the input and output labels of every transition of $\mathcal{T}$, that is, $\mathcal{T}^{-1}(x, y) = \mathcal{T}(y, x)$ for all $(x, y)$.

The *projection* of a WFST $\mathcal{T}$ is the weighted automaton denoted by $\Pi(\mathcal{T})$ obtained from $\mathcal{T}$ by omitting the input label of each transition and keeping only the output label.

The *composition* of two WFSTs $\mathcal{T}_1$ with output alphabet $\Delta$ and $\mathcal{T}_2$ with a matching input alphabet $\Delta$ is a weighted transducer defined for all $x, y$ by:

$$(\mathcal{T}_1 \circ \mathcal{T}_2)(x, y) = \bigoplus_{z \in \Delta^*} \Big( \mathcal{T}_1(x, z) \otimes \mathcal{T}_2(z, y) \Big), \tag{6}$$

where the sum runs over all strings $z$ labeling a path of $\mathcal{T}_1$ on the output side and a path of $\mathcal{T}_2$ on the input side. The worst case complexity of computing $(\mathcal{T}_1 \circ \mathcal{T}_2)$ is quadratic, that is $O(|\mathcal{T}_1||\mathcal{T}_2|)$, assuming that the $\otimes$-operation can be computed in constant time. The composition operation can also be used with WFAs by viewing a WFA as a WFST with equal input and output labels at every transition. Thus, for two WFAs $\mathcal{A}_1$ and $\mathcal{A}_2$, $(\mathcal{A}_1 \circ \mathcal{A}_2)$ is a WFA defined for all $x$ by $(\mathcal{A}_1 \circ \mathcal{A}_2)(x) = \mathcal{A}_1(x) \otimes \mathcal{A}_2(x)$.

A weighted automaton is said to be *deterministic* iff it has a unique initial state and if no two transitions leaving any state share the same input label. As for (unweighted) finite automata, there exists a *determinization* algorithm for WFAs. The algorithm returns a deterministic WFA equivalent to its input WFA (Mohri, 1997). Unlike the unweighted case, weighted determinization is not defined for all input WFAs but it can be applied to any acyclic WFA, which is the case of interest for us. When it can be applied to $\mathcal{A}$, we will denote by $\text{Det}(\mathcal{A})$ the deterministic WFA returned by determinization.

## B Sequence-to-sequence model training with rational and tropical losses

In this section, we describe how our algorithms can be incorporated into standard procedures for training modern neural network architectures for structured prediction tasks, particularly sequence-to-sequence models (Sutskever et al., 2014). Sequence-to-sequence models for structured prediction, such as RNNs and LSTMs, typically consist of an *encoder* network, which maps input data from $\mathcal{X}$ to abstract representations and a *decoder* network, which models a conditional distribution over the output space $\mathcal{Y}$. The decoder returns a $l|\Delta|$-dimensional vector of scores or *logits* $\mathbf{w}(x) = (w_{y,s}(x))$. We define $\psi(x, y^s, s)$ to be a vector of dimension $l|\Delta|$ such that the coordinate corresponding to $(y^s, s)$ is equal to one and zero otherwise. Then, setting $\Psi(x, y) = \sum_{s=1}^{l} \psi(x, y^s, s)$ defines Markovian features of order $q = 0$ as in Section 2. This allows us to use $\mathbf{w}$ and $\Psi$ to compute $F(\mathbf{w})$ in (1) along wtih its gradient $\nabla_{\mathbf{w}} F(\mathbf{w})$ in both the forward and backward pass, using techniques presented in Sections 4 and 5. In particular, $\nabla F(\mathbf{w})$ can be propagated down to lower layers of the neural network model using the chain rule.

Note that, in practice, the generation of scores from the decoder to construct these features for each $y \in \mathcal{Y}$ is expensive, and the common solution (Ranzato et al., 2015; Prabhavalkar et al., 2017) is to restrict the output vocabulary of $\Delta$ to a subset $\Delta_s$ of size $k$ at each position $s$. This is often accomplished via the beam search algorithm. In our framework, we run the beam search to construct the features automata $\mathcal{A}$, the topology of which is equal to the topology of the beam search tree.

## C  Pseudocode for Grad-Naïve

---

GRAD-NAÏVE$(x_i, y_i, \mathbf{w})$
1  $Z_{\mathbf{w}} \leftarrow \sum_{y \in \mathcal{Y}} e^{\mathsf{L}(y, y_i) + \mathbf{w} \cdot \mathbf{\Psi}(x_i, y)}$
2  **for** $(\mathbf{z}, s) \in \Delta^q \times [l]$ **do**
3      $\mathsf{Q}_{\mathbf{w}}(\mathbf{z}, s) \leftarrow \sum_{y \,:\, y^{s-q+1:s} = \mathbf{z}} e^{\mathsf{L}(y, y_i) + \mathbf{w} \cdot \mathbf{\Psi}(x_i, y)}$
4      $\mathsf{Q}_{\mathbf{w}}(\mathbf{z}, s) \leftarrow \mathsf{Q}_{\mathbf{w}}(\mathbf{z}, s) / Z_{\mathbf{w}}$

---

**Figure 6:** Computation of the key term of the gradient using the naïve direct method.