[Reviews · NeurIPS 2018]

Reviewer 1



This paper describes a tractable gradient computation for structured prediction problems, in which the authors proposed an approach to efficiently compute the expectation of feature function term that appears in the gradients using WFA. Although the proposed approach is sound, it has very restrictive assumptions: features should have the Markovian property which makes it not advantageous with respect to existing approaches. Specifically, I believe that if features are decomposable to set of local features with the Markovian property of order q and the loss is decomposable, then there exists a junction tree with tree-width of q that represents those feature. Therefore, an efficient inference in that tree is O(|\Delta|^q), which is similar to what proposed here (without considering the constant which should be a function of l ). In general, I am not convinced about the impact of the proposed algorithm since the authors compare the gradient computation with a naive algorithm that enumerates every structure; definitely dynamic programming such as inference in junction tree would end-up with very similar results based on the construction of the problem, so I would expect that sort of comparison as proof of concept instead of comparing to naive computation. Moreover, I cannot follow how the proposed approach can be used for computation of gradient in a deep neural network except the weights of the last layer. I would appreciate if the authors elaborate more on that.

Reviewer 2



Summary: This paper shows how we can compute gradients of rational and tropical structured losses using weighted finite state transducers to set up a shortest path problem. Doing so can allow us to use gradient-based training for such losses, which includes n-gram overlap and edit-distance as losses. Review: The paper presents an interesting technical contribution -- namely, a blueprint for computing gradients of certain families of losses. This could be quite useful for several applications. How long does the construction of the WFSTs take in practice? Of course, from a practical point of view, the WFSTs \mathcal{M} could be constructed when a dataset is being read because it needs to be done only once. The paper says in the introduction that if we could train with non-decomposable losses (such as F1), it might help train better models. However, this could be a bit misleading because eventually, the paper does consider decomposable losses. Related work: The direct loss minimization work of McAllester et al., 2010 and its followup work. While the proposed method is clearly different, the goals are similar. So please comment on this. While the math in the paper seems to check out, it would be useful if the paper (perhaps the appendix) provides a worked example of how \mathcal{B} is constructed for the tropical case. Right now, the reader has to infer this information from the examples — a bit more explanation would be helpful to generalize this to other new losses. Minor quibbles: - Some notation is confusing. Line 92 uses subscripts of y to indicate the i^th label. Elsewhere, y_i is used to refer to the i^th labeled example. - As an aside, perhaps it would be useful to index \mathcal{M} with the index i or the reference structure y_i because it is a function of the reference structure.

Reviewer 3



This paper proposes finite-state approaches for computing a specific loss function (eq 1) and its gradient. The approach targets improving the training-time efficiency of a finite-state transducer (FST) structured model (e.g., a chain structured CRF) under an rich family FST structured loss function (e.g., edit distance against a gold reference output string). The key contribution of the paper is the efficient computation of the family of structured losses. Finite-state methods are an obvious choice for performing these computations - it will be familiar/obvious to many NLP researchers (like myself), but perhaps less so in a machine learning venue such as NIPS/ICML. This paper goes beyond the basic construction and includes some useful discussion and correctness details. Other comments ============= - Why limit to loss functions of the form (1)? It might be useful to consider expected loss, e.g., Li & Eisner (2009) -- adding some discussion of this paper would be a useful pointer for readers. - If I'm not mistake, the loss function (1) is equivalent to soft-max margin (Gimpel & Smith, 2010; additionally Kevin Gimpel's thesis) and reward-augmented maximum likelihood (RAML; Norouzi et al., 2016). These references contain strong empirical evidence to support the choice of this loss, which are complementary to the theoretical results of reference 7 in the submission. - Have you considered a REINFORCE-style approximation as another experimental baseline (this is the approach taken in reference 26 in the submission, if I'm not mistaken)? This approach gives an unbiased (but high variance) estimate of a similar (although not the same) objective function (expected reward, which is essentially Li&Eisner'09). If I'm correct that loss function (1) is the same as RAML, then the REINFORCE-style estimator can be modified to compute this loss as well. Another reasonable approximation is to take the K-best outputs as used in Gimpel & Smith (2010), this type of approximation is very common in NLP. References ========= (Li&Eisner'09) http://www.cs.jhu.edu/~jason/papers/li+eisner.emnlp09.pdf (Gimpel & Smith, 2010) http://www.aclweb.org/anthology/N10-1112 (Norouzi et al 2016) https://papers.nips.cc/paper/6547-reward-augmented-maximum-likelihood-for-neural-structured-prediction.pdf (Kevin Gimpel's thesis) http://ttic.uchicago.edu/~kgimpel/papers/gimpel_thesis.pdf Post discussion and author response ============================ The authors have addressed my concerns in their response. These concerns were mainly about connections to other work and additional suggestion of experimental comparisons that could be made to strengthen the paper. The empirical evaluation in this paper is kind of silly: they compare an (truly) exponential-time algorithm to their polynomial-time algorithm... no surprise poly-time is much better. I suggested a randomized approximation scheme (based on Monte Carlo sampling / REINFORCE) as a baseline. They didn't bite... Re: "novelty" of the paper: Most people familiar with finite-state methods (e.g., NLP folks such as myself) would come up with this same approach. However, these authors did a reasonable good job of "packing up" their method, which will be useful to researcher. I do wish there was more "tutorial material" to help orient readers that don't know finite-state methods (since this is probably the target audience of the paper). To that end, I **really** like R2's suggestion to include a detailed worked example in the appendix. I also strongly encourage the authors to release their code in order maximize the impact of their work. I was too harsh about novelty in my initial assessment so I have increased my overall score to 7. In regards to some of R1 comments: I wanted to point out that the type of "state-space augmentation" found by finite-state methods is more fine grained than the junction-tree method used on graphical models. There are many cases where the finite-state approach yields exponentially smaller dynamic programs than the best one could do with graphical models. This is discussed in the Sum-Product networks paper (https://arxiv.org/pdf/1202.3732.pdf). The authors might want to make that connection in their paper.